# Exploring spatial variation and multilevel modeling of malaria prevalence among children aged 6-59 months based on RDT in Niger: Insights for public health decision-making

Solomon Sisay Mulugeta[1], Bezanesh Melese Masresha [2]*, Legesse Alamerie Ejigu[1]

1 Armaur Hanson Research Institute (AHRI), Addis Ababa, Ethiopia, 2 Mathiwos Wondu Foundation (MWF), Addis Ababa, Ethiopia

* bezanesh830@gmail.com

## Abstract

**Background:** Malaria is a life-threatening infectious disease caused by parasites of the genus Plasmodium transmitted through the bite of infected female *Anopheles* mosquitoes, which act as vectors of the disease. It affects approximately 219 million people globally and results in 435,000 deaths each year. Fever, chills, and exhaustion are among of the signs of this illness. If left untreated, these symptoms can develop into serious problems like anemia, respiratory distress, and even organ failure. By identifying determinants related to malaria prevalence, this study supports evidence-based national malaria prevention and control initiatives. The results help improve decision-making for malaria control efforts and guide focused public health initiatives by identifying areas with a high malaria burden.

**Methods:** Data from the 2021 Niger Malaria Indicator Survey (NMIS) is used, focusing on RDT-confirmed malaria cases in children aged 6-59 months. The dataset includes individual, household, and community-level variables, such as age, household income, education, healthcare access, and geographic coordinates. Spatial distribution of malaria prevalence is first visualized through maps and hot spot analysis to identify areas with high and low malaria rates. Random effects are incorporated to capture unobserved heterogeneity between regions and communities, allowing for more accurate estimates of malaria prevalence by adjusting for spatial clustering. Multilevel logistic regression models are applied to account for the hierarchical structure of the data. Model fit is evaluated using standard criteria (AIC, BIC and DIC), and diagnostics are performed to ensure reliability.

**Results:** 1121 (23.7%) of the 4724 children aged 6 to 59 months who were examined had positive RDT results for malaria. Malaria prevalence in Niger among children aged 6–59 months is significantly clustered (Moran's I = 0.434, p < 0.001), revealing distinct hotspots and cold spots unlikely due to chance. Model III provides a better fit for RDT prevalence among children aged 6-59 months with malaria, as

**Data availability statement:** This study utilized data from the Niger Demographic and Health Survey, which is part of the Demographic and Health Surveys (DHS) Program. The dataset is publicly accessible at https://dhsprogram.com.

**Funding:** The author(s) received no specific funding for this work.

**Competing interests:** The authors have declared that no competing interests exist.

indicated by the smallest AIC, BIC, and deviation statistics compared to other reduced models. Malaria prevalence was associated with factors, including child age, anemia levels, maternal education, the number of children sleeping under bed nets, the use of insecticide-treated nets, the number of children aged 5 and under, as well as residence and region.

**Conclusion:** The findings show that malaria prevalence among children aged 6-59 months in Niger is significantly influenced by factors such as child age, anemia levels, maternal education, and bed net usage, emphasizing the need for improved coverage of insecticide-treated nets and tailored interventions based on local conditions.

## Introduction

Malaria remains one of the great infectious killers in Africa. An estimated 300 to 500 million cases occur each year, causing 1.5 to 2.7 million deaths, primarily in children under the age of 5 years [16]. Malaria is a mosquito-transmitted infection spread by female Anopheles mosquitoes, which acts as a vector, through a bite from an infected mosquito and that affects 219 million people and causes 435 thousand deaths worldwide [20].

More than a century of international efforts and research have been made to improve malaria prevention, diagnosis, and treatment, but this high burden of disease still persists [21]. In its Global Technical Strategy for Malaria 2016-2030 (GTS), the WHO established a target in 2015 to lower the global malaria burden by 90% by 2030 in order to move closer to eradicating malaria. Nations that have already eradicated malaria must keep their status as malaria-free by preventing its reestablishment. To achieve this goal, nations must work to minimize transmission and accomplish national elimination goals within their individual countries [22].

Worldwide, approximately 2.1 billion illnesses and 11.7 million deaths were prevented between 2000 and 2022. The WHO African Region accounted for 82% of prevented cases and 94% of deaths. 233 million malaria cases were recorded by the WHO African Region in 2022, compared to 234 million in 2021. In 2022, three countries in the African Region, namely Ethiopia, Nigeria, and Uganda, experienced significant increases in malaria cases. These countries accounted for a combined increase of 1.3 million cases in Ethiopia and Nigeria, and 597,000 cases in Uganda.

The WHO African Region witnessed a high burden of malaria, with 95% of all malaria deaths (580,000) occurring in 2022. This is compared to 593,000 deaths in 2021. Regarding the "High burden to high impact" (HBHI) countries, including Burkina Faso, Cameroon, the Democratic Republic of the Congo, Ghana, Mali, Mozambique, Niger, Nigeria, Uganda, and the United Republic of Tanzania, case numbers have mostly stabilized since the pandemic. The number of deaths in these countries is returning to levels seen in 2019. In 2022, these 11 countries reported 167 million cases (67% of the global total) and 426,000 deaths (73% of the global total), similar to the 166 million cases and 430,000 deaths reported in 2021.

Young children, lacking partial immunity, are particularly vulnerable and account for most malaria deaths in the WHO African Region. Malaria remains a significant public health challenge, especially in sub-Saharan Africa, where it accounts for substantial morbidity and mortality among children. Niger is one of the countries heavily burdened by malaria, with children under five are particularly high risk due to their developing immune systems. Monitoring malaria prevalence in this vulnerable age group is essential for targeting interventions and reducing disease burden.

The use of rapid diagnostic tests (RDTs) has become a critical tool for malaria diagnosis, particularly in resource-limited settings like Niger. RDTs provide timely and accessible diagnostic results, enabling better monitoring and response to malaria outbreaks. However, understanding the spatial distribution of malaria and associated risk factors within the country is also essential, as prevalence rates can vary significantly between different regions due to factors like climate, access to healthcare, and socioeconomic conditions.

This study utilizes data from the 2023 Niger Malaria Indicator Survey (NMIS) to investigate the spatial variation of malaria prevalence among children aged 6-59 months. By applying multilevel modeling techniques, we aim to uncover key regional and individual-level factors associated with malaria prevalence, providing insights that can help guide targeted interventions. The findings from this research will contribute to a deeper understanding of malaria's spatial patterns in Niger and support public health efforts to tailor malaria control strategies to specific areas with the highest burden.

## Method of data analysis

### Study area and source of data

Niger is a landlocked country in West Africa, comprising seven regions and a capital district, with a population of approximately 23.6 million in 2021. Over 80% of its territory is covered by the Sahara Desert, resulting in an arid to semi-arid climate with annual rainfall of 100 to 700 mm. The country experiences a long dry season from October to May and a brief, irregular rainy season from June to September, which significantly affects the density and distribution of malaria vectors [10,23].

The data for this study were collected using Niger Malaria Indicator Survey (NMIS). The researchers accessed the official database of the DHS program, which is available at the website https://dhsprogram.com/. To obtain permission to use the data sets, the researchers completed online registration and application processes. During the data collection process, geographic coordinates (latitude and longitude) were recorded at the Enumeration Areas (EAs) or cluster level or clusters. This information allows for the spatial analysis and mapping of the data.

**Data management:**
For data management and analysis, several software packages were utilized. Descriptive and summary statistics were generated using Stata v14, ArcGIS V.10.8, and SaTScan V.9.6 were employed for the analysis. These software tools provide capabilities for statistical analysis, geographic information system (GIS) mapping, and spatial cluster detection.

**Ethical considerations:**
The data is downloaded from the 2021 Niger Malaria Indicator Survey (NMIS) and accessed the official database of the DHS program, which is available at the website https://dhsprogram.com/. After filling out a brief registration form, we were able to access the dataset and write the stud's title and significance on the website. With the help of ICF International, the datasets were downloaded using the above accessible website. Data were only downloaded for this study. Without EDHS' permission,the dataset was not shared with other researchers. There was no requirement to identify any household or individual respondents interviewed in the DHS because the data were treated confidentially.

### Variables in the study

**Dependent variable.** Malaria infection status among children aged 6-59 months, this variable is measured based on the results of the Rapid Diagnostic Tests (RDT) for malaria.

**Independent variables.** The Geo-statistical model utilized individual and community-level variables (Table 1).

**Table 1**. Depiction of covariates that were incorporated for the study.

| No. | variable | categories |
|---|---|---|
| 1. | Child Sex | 1=male, 2=female |
| 2. | Child age(in month) | 1=6-12, 2=13-24, 3=25-37, 4=38-50 5=50-59 |
| 3. | Region | R1=Agadez, R2=Diffa R3=Dosso, R4=Maradi, R5=Tahoua, R6=Tillaberi, R7=Zinder, R8=Niamey |
| 4. | Residence | 1=Urban, 2=Rural |
| 5. | Anemia level | 1=Anemic , 2=Non Anemic |
| 6. | Mother education level | 1=No education 2=Primary 3=Secondary and above |
| 7. | Wealth index | 1=Poorest, 2=Poor, 3=Middle, 4=Rich 5=Richest |
| 8. | Has mosquito bed nets for sleeping | 1=No, 2=Yes |
| 9. | Number of children Slept under bed net last night | 1=Null, 2=1-3, 3=4-6 |
| 10. | Number of children UN5 Slept under bed net last night | 1=None, 2=All Children 3=Somme Children, 4=No Bed Net |
| 11. | Insecticide-Treated Net (ITN) | 1=Only Treated , 2=Both 3=Not treated, 4=Did not sleep under a net |
| 12. | Number children 5 and under | 1=0-2, 2=3-5 3=6-8 |
| 13. | Number of room for sleeping | 1=1-3, 2=4-6 3=Above 7 |

**Selection bias and sample representativeness.** To minimize selection bias and maintain representativeness, sampling weights were applied, and all analyses accounted for the survey's complex design using primary sampling units and stratification variables.

## Methods

### Spatial autocorrelation and hot spot analysis.

Spatial autocorrelation, assessed through Global Moran's I, was used to examine the spatial heterogeneity. Moran's I values close to –1 indicated dispersion, while values close to +1 indicated clustering, and values around 0 suggested random distribution. A significant Moran's I value ($p < 0.05$) rejected the null hypothesis, indicating the presence of spatial autocorrelation [3]. Incremental spatial autocorrelation helped identify the distance band with the strongest clustering. Hot spot analysis, using the Getis-Ord Gi* statistic, identified significant hot or cold spots, highlighting areas of high or low adherence clustering [4]. These methods allowed for the identification of spatial patterns and determinants of adherence rates.

### Spatial interpolation.

To estimate RDT prevalence among children aged 6-59 months diagnosed with malaria in the NMIS of unsampled areas, we employed a spatial interpolation technique. Specifically, geostatistical empirical Bayesian Kriging was used with ArcGIS V.10.8 software. This method accounts for the fact that the observed semivariogram in the input data may not follow a Gaussian distribution, which is often the case in practice. Empirical Bayesian Kriging incorporates Bayes' rule to determine the weights of the new simulated semivariogram, allowing for more accurate predictions in areas without direct measurements [11].

#### Spatial scan statistics.

To identify significant clusters RDT prevalence among children aged 6-59 months diagnosed with malaria, we utilized Kuldorff's SaTScan V.9.6 software and applied Bernoulli-based model spatial scan statistics. The scanning window method was employed, where the window moved across the study area, considering RDT prevalence among children aged 6-59 months diagnosed with malaria as cases and those with adequate intake as controls. The Bernoulli model was fitted to assess the likelihood of clusters. We set the default maximum spatial cluster size to be less than 50% of the population, enabling the detection of both small and large clusters. Clusters exceeding the maximum limit and those with non-circular shapes were disregarded. The most likely clusters were determined based on p-values and log-likelihood ratio (LLR) tests, using 999 Monte Carlo replications [12].

#### Significance threshold for spatial clusters.

The statistical significance of spatial patterns and clusters was evaluated at a threshold of p < 0.05. For Moran's I and Incremental Spatial Autocorrelation, significance was assessed using 999 random permutations. Hot spot analysis with the Getis-Ord Gi* statistic was based on z-scores and corresponding p-values. For spatial scan statistics (SaTScan), statistical significance of the identified clusters was evaluated using log-likelihood ratio (LLR) tests with 999 Monte Carlo replications. Clusters with p < 0.05 were considered statistically significant.

#### Multilevel mixed-effects logistic regression analysis.

Due to the hierarchical structure of the NMIS dataset, the observations within clusters are correlated, violating the assumption of independence. The intraclass correlation (ICC) was calculated to measure the correlation within clusters [9]. The ICC formula used was ICC = VA/(VA + $\pi^2/3$), where VA represents the estimated variance in each model [17]. The proportional change in variance (PCV) was calculated to assess the total variation attributed to individual or community-level factors. PCV was computed as PCV = (VA - VB)/VA, where VA is the variance of the initial model and VB is the variance of the model with additional terms [15]. The median odds ratio (MOR) was employed to measure unexplained cluster heterogeneity and variation between clusters. MOR compares two individuals from randomly chosen clusters and is determined using the formula $MOR = \exp\left(\sqrt{2 \cdot VA \cdot 0.6745}\right) \approx \exp\left(0.95 \cdot \sqrt{VA}\right)$, VA represents the cluster-level variance. MOR is always equal to or greater than 1, with a value of 1 indicating no variation between clusters [15]. To identify factors associated with malaria prevalence among children aged 6-59 months, multilevel models were fitted. Model I, or the empty model, decomposed the total variance into individual and community-level components. Model II included individual-level factors, while Model III incorporated household-level factors. Model IV introduced community-level factors, and the final model included both individual and community-level factors. The deviance information criteria, namely Akaike's Information Criterion (AIC) and Bayesian Information Criterion (BIC), were utilized to compare the models and determine the best-fitting model [6].

## Results and discussion

### Results

Among 4724 children ages 6-59 months analyzed, 1121 (23.7%) tested positive for malaria with RDT. Out of the 4724 RDT tests conducted, 53% (594) were performed on male children and 47% (527) of female children yielded positive results for malaria. The age group with the highest positive rate was children aged 38-50 months, accounting for 29% of the total tests. This was followed by the age group of 25-37 months, with a positive rate of 25%. The age group of 13-24 months had a positive rate of 16.2%. Among the RDT tests conducted, 20.9% (985) were on individuals residing in urban areas, with a positive rate of 6.6% (74). On the other hand, 79.2% (3739) of the tests were conducted on individuals residing in rural areas, with a significantly higher positive rate of 93.4% (1047). Out of the eight regions in Niger, the region with the highest number of positive malaria cases recorded was Niamey, where 28% (313) of the RDT tests were conducted. Maradi region had a positive rate of 19.5% (219), while Tillaberi and Zinder regions both had a positive rate of 18.3% (205). These findings highlight the contrast in malaria prevalence between urban and rural areas, with higher rates

observed in rural regions. Additionally, the regional analysis reveals variations in malaria positivity rates, with Niamey, Maradi, Tillaberi, and Zinder regions having relatively higher rates compared to other regions. Among children sleeping in 1-3 rooms, 84.7% (949) tested positive for malaria. In comparison, among children sleeping in 4-6 rooms, 13.7% (153) tested positive for malaria. Regarding the use of Insecticide-Treated Nets (ITNs) among households with children under the age of five, 85.7% reported using both treated and untreated nets. Additionally, 14% reported using only treated nets. The highest positive malaria cases were among non-anemic children (54.6%), from non-educated mothers (80.4%), with the poorest economic status (24%) (Table 2).

The prevalence of positive RDT results varies widely across regions, with R8 (Niamey) showing the lowest prevalence (2.5%) and R7 (Zinder) the highest (38.3%). Overall, around 28.7% of children tested positive, which suggests significant regional differences in malaria or related infections detected by RDT.Male and female distributions are roughly equal across regions, with a slight male majority (50.7%). This balanced distribution implies that sex likely does not play a major role in RDT positivity within this age group. The age distribution varies slightly across regions, with children aged 25-37 months being the largest group (24.4%). Younger children (6-12 months) are less represented in most regions, potentially indicating lower infection exposure or differing parental behaviors in bringing younger children for testing. A vast majority of the children in the study reside in rural areas (84.3%). Urban residency is only prominent in R8 (Niamey), where 95.4% are urban residents. This rural dominance in the sample suggests that rural children may face different environmental and health-related challenges, potentially affecting their RDT results.About 12.5% of children tested are anemic, with the highest prevalence in R7 (Zinder) and R3 (Dosso). Since anemia is often associated with malnutrition and infections (e.g., malaria), regions with higher RDT positivity, like Zinder, may also show higher anemia rates. There is a high proportion of mothers with no formal education (75.4%) across regions, especially in R7 (Zinder) and R2 (Diffa). This low level of maternal education could affect awareness and preventive health practices, possibly contributing to higher RDT positivity in some regions. Wealth distribution is skewed, with the poorest households (21.1%) and the richest (17.3%) being more prominent in R1 (Agadez) and R8 (Niamey), respectively. Poorer regions generally show higher RDT positivity, suggesting a correlation between lower wealth status and higher disease prevalence. Most households (97.3%) own bed nets, and a large portion of children (82.8%) reportedly sleep under them regularly. However, regional differences, such as lower usage rates in R8 (Niamey), indicate potential gaps in effective utilization or access. Consistent bed net use aligns with lower RDT positivity in some areas, underscoring its preventive impact. Most children sleep under insecticide-treated nets (85.0%). Regions like R3 (Dosso) and R7 (Zinder) have especially high coverage of treated nets, which could help explain variations in RDT positivity across regions. Households predominantly have 0-2 children under 5 years (64.3%), but regions like R6 (Tillaberi) and R7 (Zinder) show higher counts (3-5 children). This might impact disease transmission dynamics, as larger households can present higher exposure risk.Most households have 1-3 sleeping rooms (82.6%). Crowded conditions, particularly in regions with limited rooms like R4 (Maradi), might contribute to the spread of infectious diseases, affecting RDT results (Table 3).

### Spatial autocorrelation analysis malaria prevalence among children age 6-59 months

The spatial distribution of malaria prevalence among children aged 6-59 months in Niger was clustered with Global Moran's I=0.434025 (z-score=10.471517, p value 0.001). This demonstrated the presence of spatial hotspot and cold spot clustering in Niger. With a z-score of 10.471517, there was less than a 1% chance that this high-clustered pattern was due to random chance. The tails' bright red and blue colours indicate a higher level of significance (Fig 1).

The areas highlighted in red, mostly along the southern part of Niger, indicate regions with a significantly high prevalence of malaria among children. The darkest red areas, meaning the clustering of high malaria prevalence is statistically significant in these regions. The blue areas, predominantly in the northern regions, represent locations with significantly lower prevalence rates of malaria among children. These cold spots, particularly with higher confidence levels (99% and 95%), indicate regions with fewer cases (Fig 2).

**Table 2**. **Malaria Prevalence among Children Aged 6-59 Months Diagnosed with RDT in the NMIS of 2021.** Summary of background variables (n = 4724).

| Factors | Category | Total n (%) | Prevalence of Malaria | |
|---|---|---|---|---|
| | | | Negative n (%) | Positive n (%) |
| Child Sex | Male | 2402(50.9) | 1808(38.3) | 594(12.6) |
| | Female | 2322(49.1) | 1795(38.0) | 527(11.2) |
| Child age | 6-12 | 577(12.2) | 514(10.9) | 63(1.3) |
| | 13-24 | 1078(22.8) | 896(19.0) | 182(3.9) |
| | 25-37 | 1145(24.2) | 866(18.3) | 279(5.9) |
| | 38-50 | 1074(22.7) | 750(15.9) | 324(6.9) |
| | 50-59 | 850(18.0) | 577(12.2) | 273(5.8) |
| Residence | Urban | 985(20.9) | 911(19.3) | 74(1.6) |
| | Rural | 3739(79.2) | 2692(57.0) | 1047(22.2) |
| Region | Agadez | 359(7.6) | 352(7.4) | 7(0.1) |
| | Diffa | 537(11.4) | 516(10.9) | 21(0.4) |
| | Dosso | 524(11.1) | 353(7.5) | 171(3.6) |
| | Maradi | 833(17.6) | 614(13.0) | 219(4.6) |
| | Tahoua | 636(13.5) | 467(9.9) | 169(3.6) |
| | Tillaberi | 628(13.3) | 423(9.0) | 205(4.3) |
| | Zinder | 816(17.3) | 503(10.7) | 205(4.3) |
| | Niamey | 391(8.3) | 375(7.9) | 313(6.6) |
| Anemia level | Anemic | 488(11.7) | 234(5.0) | 254(5.4) |
| | Non Anemic | 3673(88.3) | 3367(71.3) | 306(6.5) |
| Mother education level | No Education | 3002(72.4) | 2245(47.5) | 757(16.0) |
| | Primary | 651(15.7) | 519(11.0) | 132(2.8) |
| | Secondary and Above | 491(11.9) | 438(9.3) | 53(1.1) |
| Wealth index | Poorest | 1114(23.6) | 850(18.0) | 264(5.6) |
| | Poor | 831(17.6) | 581(12.3) | 250(5.3) |
| | Middle | 877(18.6) | 613(13.0) | 264(5.6) |
| | Rich | 862(18.3) | 621(13.1) | 241(5.1) |
| | Richest | 1040(22.0) | 938(19.9) | 102(2.2) |
| Has Mosquito Bed Nets | No | 234(5.0) | 209(4.4) | 25(0.5) |
| | Yes | 4490(95.1) | 3394(71.8) | 1096(23.2) |
| children Slept under Net | Null | 566(12.0) | 481(10.2) | 85(1.8) |
| | 1-3 | 3801(80.5) | 2874(60.8) | 927(19.6) |
| | 4-6 | 357(7.6) | 248(5.3) | 109(2.3) |
| children UN5 Slept under Net | No | 288(6.2) | 238(5.0) | 50(1.1) |
| | All Children | 3522(75.3) | 2651(56.1) | 871(18.4) |
| | Some | 636(13.6) | 471(10.0) | 165(3.5) |
| | No Bed Net | 234(5.0) | 209(4.4) | 25(0.5) |
| Insecticide-Treated Net | Only Treated | 829(17.6) | 673(14.3) | 156(3.3) |
| | Both | 3844(81.4) | 2883(61.0) | 961(20.3) |
| | Untreated | 3(0.1) | 2(0.0) | 1(0.1) |
| | Don't sleep under a net | 48(1.0) | 45(1.0) | 3(0.1) |
| Number children 5 and under | 0-2 | 3155(66.8) | 2468(52.2) | 687(14.5) |
| | 3-5 | 1464(31.0) | 1061(22.5) | 403(8.5) |
| | 6-8 | 105(2.2) | 74(1.6) | 31(0.7) |
| Number of room for sleeping | 1-3 | 3961(83.9) | 3012(63.8) | 949(20.1) |
| | 4-6 | 668(14.1) | 515(10.9) | 153(3.2) |
| | Above 7 | 95(2.0) | 76(1.6) | 19(0.4) |

## Spatial SaTScan analysis for prevalence of malaria among children in Niger

Spatial scan statistics revealed a total of 90 significant clusters, 40 of which were primary clusters (significant likelihood of adherence to RDT prevalence). Primary clusters were found in the Tillabéry, Dosso, and Maradi, was centred at ((13.234821 N, .808047 E)/177.87 km, with a relative risk (RR) of 2.1, meaning children in this cluster are 2.1 times more

**Table 3**. The weighted factors of RDT test results among children aged 6-59 month (n = 4869).

| Factors | Regions | | | | | | | | Total |
|---|---|---|---|---|---|---|---|---|---|
| | R1 | R2 | R3 | R4 | R5 | R6 | R7 | R8 | |
| RDT prevalence | | | | | | | | | |
| Negative | 97.9 | 96.4 | 68 | 72.3 | 72.9 | 67.9 | 61.7 | 97.5 | 71.3 |
| Positive | 2.1 | 3.7 | 32.1 | 27.7 | 27.1 | 32.1 | 38.3 | 2.5 | 28.7 |
| Male | 50.7 | 52.7 | 51.7 | 50.8 | 51.0 | 46.7 | 52.0 | 49.9 | 50.7 |
| Female | 49.3 | 47.3 | 48.3 | 49.2 | 49.0 | 53.3 | 48.0 | 50.1 | 49.3 |
| Child age | | | | | | | | | |
| 6-12 | 12.3 | 10.5 | 13.0 | 14.9 | 8.9 | 14.6 | 11.6 | 14.0 | 12.5 |
| 13-24 | 23.3 | 19.6 | 24.4 | 22.8 | 24.4 | 22.7 | 22.8 | 22.5 | 23.2 |
| 25-37 | 26.1 | 25.1 | 24.3 | 23.9 | 26.3 | 22.8 | 24.4 | 22.4 | 24.4 |
| 38-50 | 20.4 | 24.7 | 23.4 | 21.7 | 21.9 | 22.4 | 21.5 | 24.4 | 22.1 |
| 50-59 | 18.0 | 20.1 | 14.8 | 16.7 | 18.5 | 17.5 | 19.8 | 16.7 | 17.9 |
| Residence | | | | | | | | | |
| Urban | 45.8 | 25.8 | 10.1 | 15.4 | 10.9 | 6.9 | 7.2 | 95.4 | 15.7 |
| Rural | 54.2 | 74.2 | 89.9 | 84.6 | 89.1 | 93.1 | 92.6 | 4.6 | 84.3 |
| Anemia level | | | | | | | | | |
| Anemic | 11.1 | 5.5 | 15.0 | 11.1 | 13.3 | 12.3 | 15.3 | 4.2 | 12.5 |
| Non Anemic | 88.9 | 94.5 | 85.0 | 88.9 | 86.7 | 87.7 | 84.7 | 95.8 | 87.5 |
| Mother education level | | | | | | | | | |
| No Education | 59.3 | 78.4 | 70.7 | 78.4 | 77.9 | 70.8 | 84.9 | 29.9 | 75.4 |
| Primary | 17.1 | 14.2 | 15.9 | 15.5 | 14.9 | 20.4 | 9.5 | 22.3 | 15.0 |
| Secondary and Above | 23.6 | 7.4 | 13.4 | 6.1 | 7.2 | 8.9 | 5.7 | 47.8 | 9.6 |
| Wealth index | | | | | | | | | |
| Poorest | 21.4 | 41.7 | 23.9 | 17.4 | 14.9 | 28.1 | 27.1 | 1.1 | 21.1 |
| Poor | 12.4 | 8.1 | 20.8 | 14.6 | 26.7 | 18.1 | 26.1 | 0.9 | 20.0 |
| Middle | 9.6 | 9.4 | 23.9 | 25.2 | 25.8 | 17.7 | 22.7 | 0.6 | 21.7 |
| Rich | 10.7 | 14.4 | 19.1 | 27.7 | 19.1 | 25.2 | 14.5 | 4.8 | 19.9 |
| Richest | 45.9 | 26.5 | 12.4 | 15.2 | 13.6 | 11.0 | 9.7 | 92.6 | 17.3 |
| Has Bed Nets | | | | | | | | | |
| No | 5.9 | 18.1 | 2.1 | 2.2 | 1.3 | 4.6 | 0.4 | 8.4 | 2.7 |
| Yes | 94.1 | 81.9 | 97.9 | 97.8 | 98.7 | 95.4 | 99.6 | 91.6 | 97.3 |
| children Slept under Net | | | | | | | | | |
| Null | 18.2 | 20.4 | 10.0 | 5.4 | 7.8 | 10.8 | 4.9 | 33.2 | 8.8 |
| 1-3 | 80.0 | 71.6 | 85.3 | 84.8 | 86.4 | 84.1 | 81.8 | 61.5 | 82.8 |
| 4-6 | 1.89 | 8.0 | 4.8 | 9.7 | 5.8 | 5.2 | 13.3 | 5.3 | 8.4 |
| CUN5 Slept under Net | | | | | | | | | |
| No | 12.4 | 2.3 | 6.6 | 3.0 | 4.4 | 4.7 | 3.7 | 22.9 | 5.1 |
| All Children | 67.1 | 64.8 | 79.9 | 78.7 | 84.6 | 73.0 | 84.0 | 51.8 | 78.7 |
| Some | 14.7 | 14.8 | 11.4 | 16.1 | 9.7 | 17.7 | 11.9 | 16.6 | 13.6 |
| No Bed Net | 5.9 | 18.1 | 2.1 | 2.3 | 1.4 | 4.7 | 0.4 | 8.6 | 2.7 |
| Insecticide Treated Net | | | | | | | | | |
| Only Treated | 24.9 | 25.4 | 15.2 | 12.2 | 13.0 | 17.3 | 8.5 | 40.2 | 14.3 |
| Both | 73.8 | 72.1 | 84.3 | 87.6 | 87.0 | 81.3 | 90.8 | 56.1 | 85.0 |
| Not treated | 0.0 | 0.0 | 0.0 | 0.1 | 0.0 | 0.0 | 0.0 | 0.6 | 0.1 |
| Not under net | 1.4 | 2.6 | 0.5 | 0.1 | 0.0 | 1.4 | 0.7 | 3.2 | 0.7 |
| | Regions | | | | | | | | |
| Factors | R1 | R2 | R3 | R4 | R5 | R6 | R7 | R8 | Total |
| No. children 5 and under | | | | | | | | | |
| 0-2 | 73.4 | 68.6 | 73.3 | 55.4 | 75.0 | 64.9 | 57.8 | 74.2 | 64.3 |
| 3-5 | 26.6 | 30.8 | 26.4 | 40.2 | 25.0 | 33.5 | 37.8 | 22.6 | 33.2 |
| 6-8 | 0.0 | 0.7 | 0.3 | 4.4 | 0.0 | 1.5 | 4.4 | 3.3 | 2.6 |
| Number of room for sleeping | | | | | | | | | |
| 1-3 | 87.9 | 90.5 | 85.4 | 76.3 | 87.5 | 85.3 | 82.3 | 75.7 | 82.6 |
| 4-6 | 10.5 | 7.8 | 13.9 | 20.3 | 11.6 | 14.6 | 15.7 | 16.5 | 15.4 |
| Above 7 | 1.7 | 1.6 | 0.8 | 3.4 | 0.9 | 0.1 | 2.0 | 7.7 | 2.0 |

R1=Agadez, R2=Diffa, R3=Dosso, R4=Maradi.
R5= Tahoua, R6=Tillaberi, R7=Zinder, R8=Niamey.

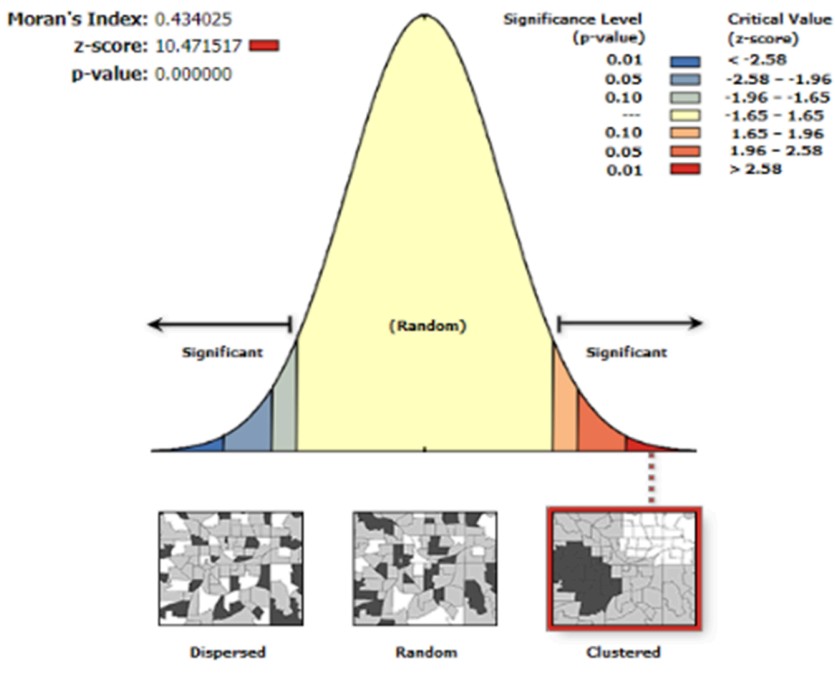

Given the z-score of 10.4715166739, there is a less than 1% likelihood that this clustered pattern could be the result of random chance.

**Fig 1**. **Spatial autocorrelation analysis malaria prevalence among children age 6-59 months in Niger, 2021 Niger malaria indicator Survey.**
Given the Z- score of 10.47 there is a less than 1% likelihood that this clustered pattern could be the result of random chance.

likely to adhere to RDT prevalence than those outside the cluster and an LLR of 91.2 with p value <0.0001. It was discovered that children who lived within the most likely cluster were 91% more likely to adhere to RDT than children who did not live within the spatial window (Table 4 and (Fig 3)).

The light orange areas reflect a moderate prevalence of malaria, while the yellow parts exhibit the lowest prevalence. The areas with the highest malaria prevalence were identified as being dark orange, whereas those with a significant malaria load were shown in red. The highest prevalence of malaria is found in the southwestern regions, which include Tillabéry, Dosso, and Maradi (Fig 4)).

The findings of the random logistic regression analysis are summarized in Table 4. The empty model (Model I) shows that there are discrepancies in the prevalence of RDT for children. The ICC within the null model revealed that community-level variability accounted for 36.4% of the variability in RDT prevalence. Additionally, the null model's MOR of 3.6 indicates that there was variation in adherence to RDT prevalence between clusters. Furthermore, the PCV of the whole model (model IV) revealed that individual and community factors accounted for roughly 50 percent of the difference in prevalence of RDT among children aged 6-59 months diagnosed with malaria. The unexplained community variance in prevalence of RDT for children was decreased to MOR=2 when all factors were incorporated to the null model. When all factors are taken into account, the effect of clustering remains statistically significant in the overall model (Table 5).

Older children (especially those 25 months and older) have significantly higher odds of testing positive for malaria, with the odds increasing as age increases.

Children in rural areas have significantly higher odds of testing positive for malaria compared to those in urban areas.

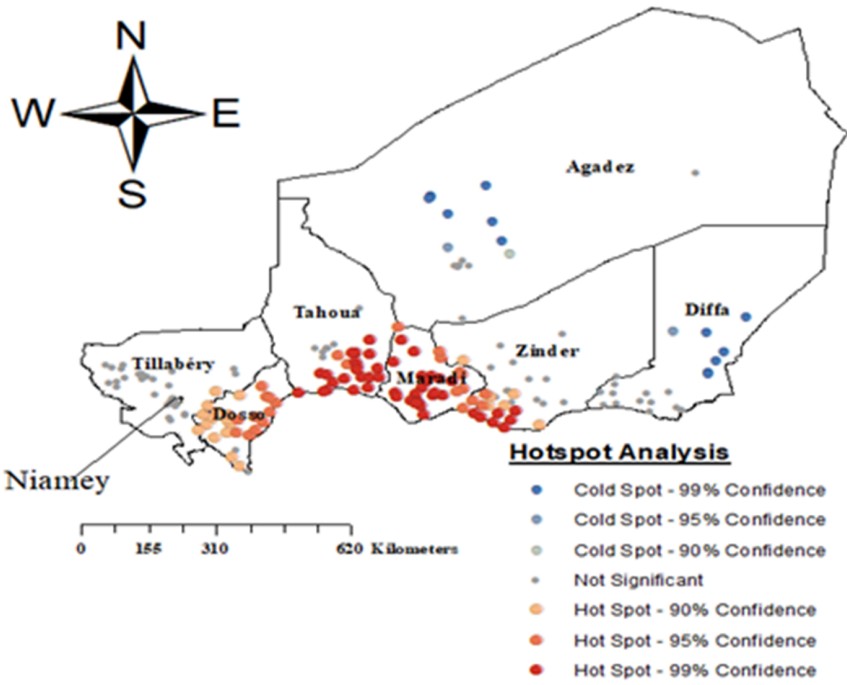

**Fig 2**. **Hot spot analysis of prevalence of malaria among children in Niger, Map created by the authors using ArcGIS software and cluster data from the 2021 Niger Malaria Indicator Survey (NMIS),** DHS Program. Copyright rests with the authors and is distributed under the CC BY 4.0 license.

Anemia is strongly associated with malaria prevalence. Children who are non-anemic have a much higher likelihood of testing positive for malaria.

Maternal education, particularly primary education, is associated with a slight increase in the odds of a child testing positive for malaria.

The AOR is 0.5 (CI: 0.26–0.90) in Model I, indicating that children using treated nets are less likely to test positive for malaria. This result is even more pronounced in Model II (AOR: 0.5, CI: 0.27–0.91).

The AOR is 1.4 (CI: 1.04–1.89) in Model I and remains at 1.4 (CI: 1.03–1.87) in Model II, suggesting that households with 3-5 children under 5 years have higher odds of a child testing positive for malaria.

Significant regional variations are observed, with areas like Zinder, Dosso, Tillaberi, and Maradi showing extremely high odds of testing positive for malaria.

Zinder: The AOR is 23.5 (CI: 6.19–89.11) in the null model, and increases to 151.1 (CI: 9.82–2326.02) in Model II, indicating very high odds of malaria positivity. Other regions such as Dosso, Tillaberi, and Maradi also show increased odds of testing positive, with AOR values as high as 115.6 and 54.0, respectively and Niamey shows the smallest odds ratio of 2.3 (Tables 6 and 7).

## Discussion

The results of this study shed important light on the factors and geographic variation in the prevalence of malaria in Niger's 6-59-month-old population. Significant clustering of malaria prevalence is revealed by the results, indicating clear geographical differences impacted by behavioral, socioeconomic, and environmental factors. With a higher incidence in particular areas like Zinder, Dosso, Maradi, and Tillaberi, the spatial analysis demonstrates that malaria prevalence is not dispersed randomly but rather shows considerable clustering. These results are consistent with earlier studies showing

**Table 4**. Significant SaTScan clusters of adherence to RDT among children aged 6-59 moths in Niger, 2021 Niger malaria indicator Survey.

| Most likely clusters | clusters detected | Population(n) | Cases(n) | RR | LL | coordinates | P-Value |
|---|---|---|---|---|---|---|---|
| Primary | 169, 174, 173, 167, 170, 160, 171, 168, 176, 172, 175, 166, 165,178,177, 186,185, 90, 179, 76, 161, 162, 164, 91, 93, 158, 181, 69, 87,92, 159, 86, 182, 163, 89, 79, 88, 180, 81, 82 | 1113 | 438 | 2.1 | 91.2 | (13.234821 N, .808047 E)/177.87 km | <0.0001 |
| Secondary[a] | 62, 60, 61, 63, 47, 57, 66, 54, 59, 52, 65, 55, 53, 67, 58, 44, 56,43, 68, 45, 46, 48,135,50, 141, 64, 49, 138, 143, 142, 129, 51,137, 100 | 728 | 259 | 1.7 | 31.0 | (11.749061 N, .650199 E)/248.58 km | <0.0001 |
| Secondary[b] | 113, 115, 106, 104, 114, 116, 118, | 170 | 75 | 1.9 | 17.7 | (14.147479 N, .971417 E)/40.02 km | 0.000012 |
| Secondary[c] | 144, 147, 145, 149, 155, 133, 150, 148, 134 | 164 | 69 | 1.8 | 13.9 | (14.138878 N, .090793 E)/38.71 km | 0.00029 |

that human activities that impact mosquito breeding grounds, vector density, and regional climate are factors that impact the spread of malaria.

The multilevel modeling approach used in this study shows the complex interplay of individual, household, and community-level factors associated with malaria prevalence. At the individual level, child age was a significant predictor, with older children exhibiting higher odds of testing positive for malaria. This trend may be attributed to increased exposure to mosquito bites as children grow older and become more mobile. Additionally, children from rural areas had significantly higher odds of malaria infection compared to their urban counterparts [1,14] and with non-anemic children showing a higher likelihood of malaria infection in which the findings align with existing evidences [2,7].

Similar to others one important factor that has been identified as influencing the prevalence of malaria was maternal education. Malaria prevalence was higher in children of mothers with lower levels of education, indicating that maternal awareness and knowledge are important factors in malaria prevention strategies. Mothers who have received education are more likely to use bed nets and seek treatment for their children in a timely manner, which lowers the chance of contracting malaria [5,13,19].

Agreeing with insights from areas [24], household factors, including the number of children sleeping under insecticide-treated nets (ITNs), also played a role in malaria prevalence. The results indicate that households with a higher number of children under bed nets had lower odds of malaria infection, reinforcing the protective effect of ITN usage. However, despite high bed net ownership rates, disparities in effective utilization remain a concern, particularly in rural.

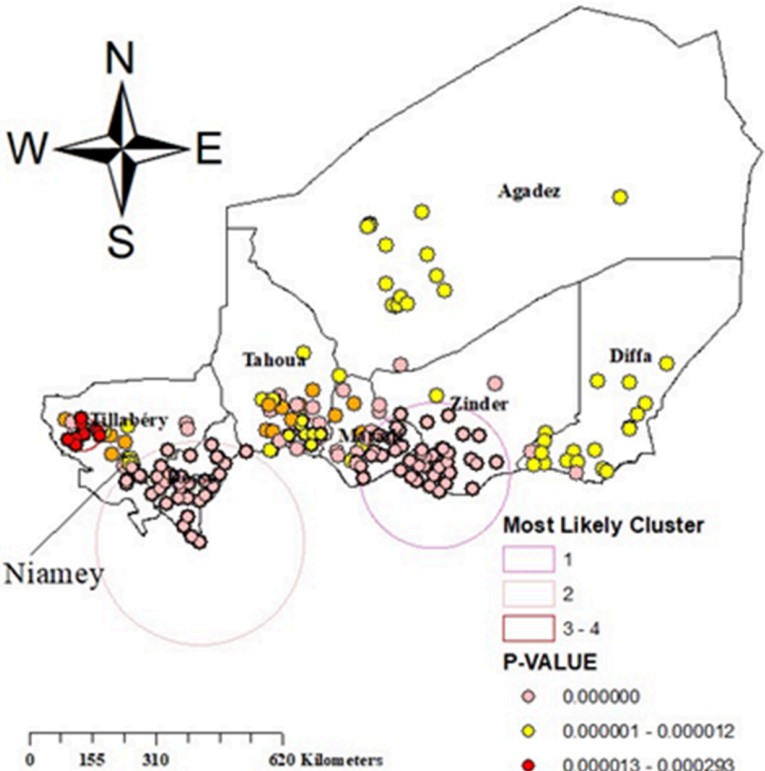

**Fig 3. SaTScan analysis for prevalence of malaria among children in Niger, Map created by the authors using ArcGIS software and cluster data from the 2021 Niger Malaria Indicator Survey (NMIS), DHS Program.** Copyright rests with the authors and is distributed under the CC BY 4.0 license.

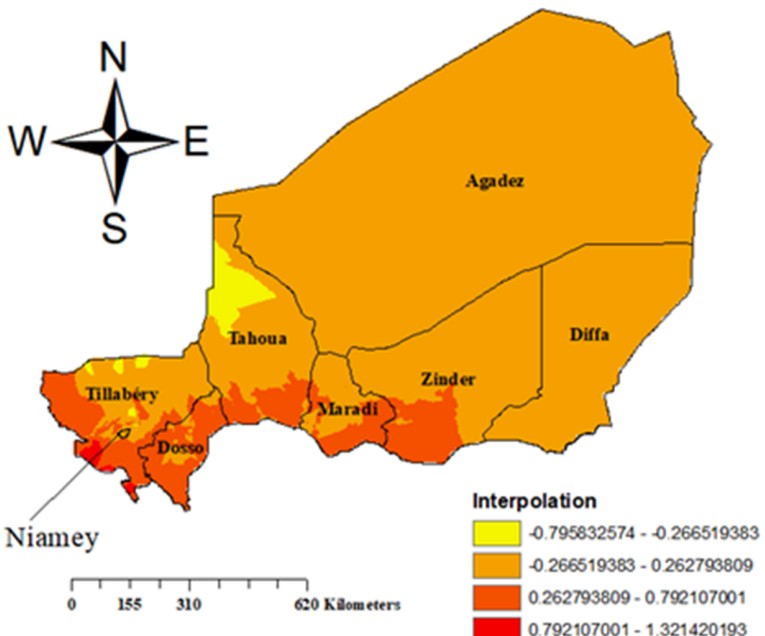

**Fig 4. Interpolation of prevalence of malaria among children in Niger, Map created by the authors using ArcGIS software and cluster data from the 2021 Niger Malaria Indicator Survey (NMIS), DHS Program.** Copyright rests with the authors and is distributed under the CC BY 4.0 license.

**Table 5**. Interpolation of prevalence of malaria among children in Niger, 2021 Niger malaria indicator Survey.

| Measure of variation | Model | | | |
| --- | --- | --- | --- | --- |
| | Null model | Model I | Model II | Model III |
| Community variance (SE) | 1.9(0.35) | 1.4(0.28) | 1.2(0.21) | 0.6(0.10) |
| ICC(%) | 36.4 | 29.7 | 27.4 | 15.4 |
| PCV(%) | 1 | 26.3 | 14.3 | 50.0 |
| MOR | 3.6 | 3.1 | 2.8 | 2.0 |
| Model fit statistics | | | | |
| -2*LL(DIC) | 5358.4 | 2057.4 | 5210.4 | 1995.4 |
| AIC | 5362.3 | 2101.3 | 5231.2 | 2055.5 |
| BIC | 5375.3 | 2237.8 | 5295.8 | 2241.6 |

AIC, Akaike's information criterion; BIC, Bayesian information criterion; DIC, deviance information criteria; ICC, intraclass correlation; MOR, median OR; PCV, proportional change in variance.

Our study found that non-anemic children were more likely to test positive for malaria compared to anemic children, which appears counter-intuitive. This may be explained by the multifactorial nature of anemia, which can result from nutritional deficiencies, hemoglobinopathies, or other infections besides malaria. In addition, many malaria infections may be asymptomatic or in early stages before hemoglobin decline occurs. Evidence also suggests that iron deficiency and some hemoglobin disorders may reduce malaria parasite growth, offering partial protection [8,18].

The spatial scan statistics identified significant malaria clusters, with children residing within high-risk clusters being more than twice as likely to test positive for malaria compared to those outside these clusters. These findings suggest the need for geographically targeted interventions, such as indoor residual spraying and intensified vector control measures in high-prevalence areas. By prioritizing regions with the highest burden, malaria control programs can achieve more efficient resource allocation and improved public health outcomes.

**Limitations and future directions.**

Regarding the need for a deeper discussion on regional disparities and the limitations of RDTs as a diagnostic tool, while these factors are critical for a comprehensive analysis of malaria prevalence, our current study's scope is constrained by the available data. The existing dataset does not contain sufficient detail on regional infrastructure, socioeconomic factors, or data from alternative diagnostic methods to support a robust and data-driven discussion on these points. Therefore, we recommend that future research in this area be designed to collect more granular data that would allow for a more thorough exploration of these important structural factors. This would provide valuable insights into the underlying drivers of the regional disparities observed in this study and offer a more complete picture of malaria epidemiology in the region.

## Conclusion

This study highlights that malaria among under-five children in Niger remains deeply entrenched not only as a biomedical issue but as a structural and regulatory challenge. Maternal education, bed net use, anemia levels, and child age all had a substantial impact on the prevalence of malaria in Niger among children aged 6 to 59 months, according to the study. The findings emphasize the need of increasing insecticide-treated net coverage and putting customized interventions into place according to local conditions. The study also highlights the need of using geographical analysis to pinpoint high-risk locations, which can improve malaria control efforts and the efficient use of resources. The impact of malaria on the most vulnerable people can also be better understood by knowing the prevalence of the disease in particular age groups. To reduce the burden, malaria control must shift from broad policy statements to region-specific, enforceable strategies with stronger local governance and monitoring.

**Table 6**. Multivariate multilevel logistic regression analysis of child and community level factors associated with prevalence of RDT among children aged 6-59 months diagnosed with malaria in the NMIS of 2021.

| variables | Model | | | | |
| --- | --- | --- | --- | --- | --- |
| | Null model | Model I | | Model II | Model III |
| | | AOR (95% CI) | | | AOR(95% CI) |
| Child sex | | | | | |
| Male | | 1 | | | 1 |
| Female | | 0.9(0.68-1.11) | | | 0.9(0.68-1.12) |
| Child age | | | | | |
| 6-12 | | 1 | | | 1 |
| 13-24 | | 1.7(1.03-2.85)* | | | 1.7(1.04-2.88)** |
| 25-37 | | 5.4(3.31-8.87)*** | | | 5.6(3.4- 9.15)*** |
| 38-50 | | 6.6(4.01-11.01)*** | | | 6.9(4.17-11.51)*** |
| 50-59 | | 8.5(5.06-14.33)*** | | | 8.8(5.22-14.86)*** |
| Residence | | | | | |
| Urban | | 1 | | | 1 |
| Rural | | 3.2(1.96-5.12)*** | | | 5.1(1.76-14.64)*** |
| Anemia level | | | | | |
| Anemic | | 1 | | | 1 |
| Non Anemic | | 27.2(19.51-37.88)*** | | | 27.5(19.72-38.4.0)*** |
| Mother education level | | | | | |
| No Education | | 1 | | | 1 |
| Primary | | 1.6(1.10-2.26)* | | | 1.7(1.19-2.45)** |
| Secondary | | | | | |
| and Above | | 1.1(0.60-1.84) | | | 1.2(0.67- 2.09) |
| Wealth index | | | | | |
| Poorest | | 1 | | | 1 |
| Poor | | 0.9(0.59-1.33) | | | 0.9(0.59-1.33) |
| Middle | | 0.9(0.62-1.41) | | | 0.9(0.63-1.43) |
| Rich | | 0.6(0.37-0.93)* | | | 0.7(0.41-1.04) |
| Richest | | 0.3(0.15-0.57)*** | | | 0.8(0.35-1.63) |
| children Slept under Net | | | | | |
| Null | | 1 | | | 1 |
| 1-3 | | 1.6( 0.73-3.57) | | | 1.5(0.65-3.24) |
| 4-6 | | 3.6( 1.44-9.78)** | | | 3.3(1.28-8.76)** |
| CUN5 Slept under Net | | | | | |
| No | | 1 | | | 1 |
| All Children | | 1.4(0.83-2.24) | | | 1.3(0.79-2.14) |
| No Bed Net | | 1.1(0.37-3.03) | | | 1.3(0.43-0.68) |
| Insecticide Treated Net | | | | | |
| Not under net | | 0.9(0.68-1.11) | | | 0.9(0.68-1.12) |
| Only Treated | | 0.5(0.26-0.90)* | | | 0.5(0.27-0.91)** |
| Not treated | | 0.1(0.00-0.87)* | | | 0.1(0.00-0.85)** |
| No. children 5 and under | | | | | |
| 0-2 | | 1 | | | 1 |
| 3-5 | | 1.4(1.04-1.89)* | | | 1.4(1.03-1.87)** |
| 6-8 | | 0.7(0.30-1.71) | | | 0.7(0.27-1.56) |
| Region | | | | | |
| Agadez | | 1 | | | 1 |
| Diffa | | 1.4(0.27-7.59) | | | 10.9(0.52-230.17) |
| Dosso | | 19.8(5.11-76.49)*** | | | 115.6(7.35-1816.04)** |
| Maradi | | 15.1(3.98-57.42)*** | | | 54.0(3.49-835.92)** |
| Tahoua | | 13.4(3.52-51.05)*** | | | 38.2(2.46-592.79)** |
| Tillaberi | | 19.3(5.02-74.18)*** | | | 78.3( 5.00-1226.34)** |
| Zinder | | 23.5(6.19-89.11)*** | | | 151.1(9.82- 2326.02)*** |
| Niamey | | 2.3(0.45-11.5) | | | 34.1(1.64-709.79)*** |

**Table 7**. Key Findings and Implications of Malaria Prevalence Study among Children Aged 6-59 Months in Niger (NMIS 2021).

| Variable | Key Findings | Implications |
|---|---|---|
| Overall Prevalence | 23.7% (1,121 out of 4,724) of children aged 6-59 months tested positive for malaria (RDT). | Malaria remains a major public health concern among under-five children in Niger, requiring sustained prevention and control efforts. |
| Spatial Distribution | Significant spatial clustering (Moran's I = 0.434, $p < 0.001$) identified malaria hotspots mainly in southern regions (Zinder, Dosso, Maradi, Tillaberi). | Interventions should be geographically targeted, prioritizing high-risk southern zones for intensified vector control and resource allocation. |
| Residence | Rural children were 3-5 times more likely to test positive than urban children (AOR = 5.1, $p < 0.001$). | Scale up access to malaria prevention and diagnosis in rural communities, including outreach and improved health service access. |
| Bed Net Use | Use of insecticide-treated nets significantly reduced malaria risk (AOR = 0.5, $p < 0.01$). | Reinforce ITN distribution and utilization campaigns, focusing on effective use rather than ownership alone. |
| Household Characteristics | Households with 3-5 children under five had higher odds of malaria (AOR = 1.4, $p < 0.05$). | Promote family-centered prevention education and ensure adequate bed net coverage per household. |
| Regional Effects | Zinder (AOR = 151.1), Dosso (AOR = 115.6), Tillaberi (AOR = 78.3), and Maradi (AOR = 54.0) showed extremely high malaria odds. | Urgent need for region-specific malaria control programs and localized monitoring systems. |
| Model Fit & Variation | Multilevel modeling revealed that 36% of variation in malaria prevalence was due to community-level differences (ICC = 36.4%). | Community-level interventions (e.g., sanitation, housing improvement) are critical alongside individual-level efforts. |

## List of acronyms

- AIC Akaike's Information Criterion
- BIC Bayesian Information Criterion
- EA Enumeration Areas
- GIS Geographic Information System
- HBHI High Burden to High Impact
- ICC Intraclass Correlation
- LLR Log-Likelihood Ratio
- MOR Median Odds Ratio
- NMIS Niger Malaria Indicator Survey
- PCV Proportional Change in Variance
- RDT Rapid Diagnostic Tests
- SaTScan Spatial and Temporal Scan Statistics.

## Acknowledgments

The authors of this article would like to thank the Niger Malaria Indicator Survey With the official database of the DHS program for making the data available.

## Author contributions

**Conceptualization:** Solomon Sisay Mulugeta, Bezanesh Melese Masresha.

**Data curation:** Solomon Sisay Mulugeta.

**Formal analysis:** Solomon Sisay Mulugeta.

**Methodology:** Solomon Sisay Mulugeta, Bezanesh Melese Masresha, Legesse Alamerie Ejigu.

**Software:** Solomon Sisay Mulugeta.

**Validation:** Legesse Alamerie Ejigu.

**Visualization:** Legesse Alamerie Ejigu.

**Writing – original draft:** Bezanesh Melese Masresha.

**Writing – review & editing:** Bezanesh Melese Masresha.

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
