## [Decision Letter · Decision Letter 0]

9 Sep 2025

PONE-D-25-18518Exploring Spatial Variation and Multilevel Modeling of Malaria Prevalence Among Children Aged 6-59 Months Based on RDT in Niger: Findings from the 2021 NMISPLOS ONE

Dear Dr.
Masresha,

Thank you for submitting your manuscript to PLoS ONE. After careful consideration, we felt that your study has the potential to be published if it is revised to address specific topics raised by the reviewers. According to the reviewers, there are some specific areas where further improvements would be of substantial benefit to the readers.  For example, some inherent limitations of the study design should be included; in specially, the limitations of observational studies should be acknowledged.  Also, typographical and grammatical errors throughout the manuscript should be adjust. For your guidance, a copy of the reviewers' comments was included below. Finally, please be sure to follow the PLOS ONE policies for publishing observational studies. For studies involving clinical data, use the STROBE checklist, and for studies with routinely collected data, use the RECORD checklist.

We look forward to receiving your revised manuscript.

Kind regards,

Luzia H Carvalho, Ph.D.

Academic Editor

PLOS ONE

2. Please include a separate caption for each figure in your manuscript.

3. We note that Figures 2, 3, and 4 in your submission contain [map/satellite] images which may be copyrighted. All PLOS content is published under the Creative Commons Attribution License (CC BY 4.0), which means that the manuscript, images, and Supporting Information files will be freely available online, and any third party is permitted to access, download, copy, distribute, and use these materials in any way, even commercially, with proper attribution. For these reasons, we cannot publish previously copyrighted maps or satellite images created using proprietary data, such as Google software (Google Maps, Street View, and Earth). For more information, see our copyright guidelines: http://journals.plos.org/plosone/s/licenses-and-copyright.

1. You may seek permission from the original copyright holder of Figures 2, 3, and 4 to publish the content specifically under the CC BY 4.0 license. 

Additional Editor Comments (if provided):

Reviewer #1:

Reviewer #2:

Reviewers' comments:

Reviewer's Responses to Questions

**Comments to the Author**

1. Is the manuscript technically sound, and do the data support the conclusions?

Reviewer #1: Yes

Reviewer #2: Yes

2. Has the statistical analysis been performed appropriately and rigorously?

Reviewer #1: Yes

Reviewer #2: Yes

3. Have the authors made all data underlying the findings in their manuscript fully available?

Reviewer #1: Yes

Reviewer #2: Yes

4. Is the manuscript presented in an intelligible fashion and written in standard English?

Reviewer #1: Yes

Reviewer #2: No

5. Review Comments to the Author

Reviewer #1: 1. Scientific relevance and originality

The study addresses a relevant and major public health issue in sub-Saharan Africa: the prevalence of malaria in children under five. The combined approach of spatial analysis and multilevel modeling is relevant and well justified. The originality lies in the use of NMIS 2021 data to produce a fine mapping of at-risk areas and identify multi-level determinants.

Assessment: Relevant subject, innovative methodological approach for the Niger context.

2. Methodology

The article uses robust methods:

- Spatial analysis (Moran's I, Getis-Ord Gi*, SaTScan)

- Geostatistical interpolation (Empirical Bayesian Kriging)

- Multilevel modeling (mixed-effects logistics)

The models are well described, with goodness-of-fit indicators (AIC, BIC, DIC, ICC, MOR, PCV). Stratification of variables at different levels (individual, household, community) is rigorous.

Suggestions:

- Clarify certain methodological choices, in particular the significance threshold for spatial clusters.

- Further justify the interpretation of certain counter-intuitive associations (e.g. non-anemia associated with higher prevalence).

3. Results

The results are clearly presented, with detailed tables and relevant geographical figures. The identification of high-risk clusters in the Zinder, Dosso, Maradi and Tillabéri regions is particularly useful for decision-makers.

Strengths:

- Good visualization of spatial data.

- Well-structured multi-level analysis.

Areas for improvement:

- Summarize key results more in the text to avoid number overload.

- Standardize percentages and headcounts in tables.

4. Discussion and interpretation

The discussion is well grounded in the existing literature. The authors interpret the results with caution and highlight the implications for public health policy.

Recommendations:

- Deepen discussion of regional disparities and structural factors (access to care, infrastructure).

- Incorporate consideration of the limitations of RDTs as a diagnostic tool (sensitivity, specificity).

5. Conclusion

The conclusion is consistent with the results and highlights concrete recommendations: increased coverage of impregnated mosquito nets, targeted interventions according to risk areas, and the importance of maternal education.

Assessment: Relevant conclusion, well aligned with study objectives.

6. Ethics, data availability and conflicts of interest

- The authors state that the data come from the DHS Program and are publicly available.

- No conflicts of interest declared.

- No direct collection of human data: ethics respected.

Reviewer #2: In this manuscript, Mulugeta et al. present the results of an epidemiological study that explored spatial variation and fitted a model for malaria prevalence among children aged 6 to 59 months in Niger. Although not original, the study is of great importance because it focused on a high-risk age group with high mortality for malaria in Africa, and it employed correct, robust, and up-to-date statistical methodologies to achieve the proposed objectives.

The statistical model proposed in the study is well explained and described. Authors utilized appropriate statistical tools for model fitting, such as the AIC, BIC, and DIC criteria, as well as to explain cluster heterogeneity in spatial analysis (MOR and PCV). The multilevel multivariate logistic regression is well-structured and clearly presented.

The manuscript is well written and clearly presented. Discussion and Conclusion were consistent with the study objectives and the results obtained. Moreover, the authors highlight important recommendations for incorporation into the malaria control strategy targeting the studied age group.

Although the authors were cautious in stating the study’s conclusions, some inherent limitations of the study design were omitted. Among the potential limitations are those inherent to the use of secondary data, which are subject to issues of quality and coverage, as well as the expected underestimation of new malaria cases, arising both from the existence of undiagnosed cases (asymptomatic or not reported in the information system). Another limitation that cannot be omitted is the low sensitivity of rapid diagnostic tests (RDTs) for malaria cases with low parasitemia.

Additionally, the authors should correct various typographical and grammatical errors throughout the manuscript to improve the clarity and professionalism of the scientific writing. Special attention to:

1. Missing space after period at the end of a sentence, occurring in several sentences of the text.

2. "known as a vector of infection" could be simplified to "which acts as a vector".

3. "a bite from an infected insect". It is better as "a bite from an infected mosquito".

4. "This is compared to 593,000 cases in 2021" refers to deaths but uses "cases" erroneously - should be "deaths".

5. "children under five at particularly high risk" should be "children under five years old are at particularly high risk".

6. "[?, 12, 28]" The question mark indicates a missing or placeholder reference; it needs to be replaced with the proper citation number.

7. "Niger malaria indicator survey (NMIS)" - "Niger Malaria Indicator Survey (NMIS)" (capitalize proper nouns).

8. "geographic coordinates (longitude and latitude)" - Standard order is "latitude and longitude."

9. "Enumeration Areas (EAs) or cluster level" - Should be "Enumeration Areas (EAs) or cluster level" or "clusters".

10. "Correct capitalization and style: "Stata v14".

11. Missing space after comma: "ICF International, the datasets".

12. "accessible web-site" — "website" is one word; hyphen unnecessary.

13. "write the studys title and signifi-cance" — Should be "study’s title and significance" with an apostrophe; hyphenation in "signifi-cance" should be corrected to "significance".

14. "Independent Variables The Geo-statistical model utilized individual and community-level variables 5.” – What is this 5?

15. Hyphenation & spacing: Several places require adding spaces after punctuation marks before continuing, e.g., after commas and periods.

16. Consider defining acronyms upon first usage if not done elsewhere, e.g., "RDT," "SaTScan," "ICC," "VA," "PCV," etc.

17. “but "47% (527)" says "female child yielded" where "child" should be plural "children".

18. "Out of the 4724 RDT tests conducted, 53% (594) were performed on male children, and 47% (527) of female child yielded positive results for malaria." Should be: "...and 47% (527) of female children yielded positive.

19. "The tails bright red and blue colours indicate a higher level of significance (Fig 1)." - Remove extra space before parentheses. "Globally Morans I=0.434025 (z-score=10.471517, p value 0.001)." Should be: "Global Moran's I = 0.434 (z-score = 10.47, p = 0.001)." Use apostrophe in "Moran's".

There are several other typographical errors not mentioned above. The authors are advised to make the necessary corrections.

6. PLOS authors have the option to publish the peer review history of their article (what does this mean?). If published, this will include your full peer review and any attached files.

Reviewer #1: **Yes: **Dr Bougouma Edith Christiane / Bougouma.cedith@gmail.com

Reviewer #2: No

---

## [Author Response · Author response to Decision Letter 1]

23 Sep 2025

Review Comments to the Author

Reviewer #1:

2. Methodology

Suggestions:

- Clarify certain methodological choices, in particular the significance threshold for spatial clusters.

- Further justify the interpretation of certain counter-intuitive associations (e.g. non-anemia associated with higher prevalence).

Response

We have made revision with significance threshold details in separate section and define how significance was assessed in Moran’s I, Gi*, and SaTScan.

We appreciate the reviewer’s observation regarding the counter-intuitive finding. Several possible explanations may account for our result. We have revised the discussion section to include these points and provide a more balanced interpretation.

3. Results

Areas for improvement:

- Summarize key results more in the text to avoid number overload.

- Standardize percentages and headcounts in tables.

Response

-Thank you for your valuable feedback on the table's structure. As per your recommendation, we have revised the table to ensure all data is presented in a standardized format. Specifically, we have implemented a uniform basis for all percentage calculations, using a single, consistent denominator across the entire table.

-We appreciate the feedback regarding the need to summarize our key results more concisely to avoid 'number overload.' Given the nature of our spatial modeling, which generates extensive data, we have carefully selected only the most relevant and essential numerical figures for inclusion in the main text. Presenting these specific values is crucial, as further removal would render the analysis less informative and compromise the integrity of our findings. Therefore, we have opted to prioritize a comprehensive presentation of our results over a text-heavy narrative, ensuring the reader has access to the core data supporting our conclusions.

4. Discussion and interpretation

Recommendations:

- Deepen discussion of regional disparities and structural factors (access to care, infrastructure).

- Incorporate consideration of the limitations of RDTs as a diagnostic tool (sensitivity, specificity).

Response

While we agree that these points are crucial for a comprehensive study, we cannot introduce information because of the available data.

Having your Recommendations we added a section of Limitations and Future Directions to the discussion that acknowledges these limitations as areas for future research.

5. Review Comments to the Author

Reviewer #2:

Reviewer #2:

Additionally, the authors should correct various typographical and grammatical errors throughout the manuscript to improve the clarity and professionalism of the scientific writing. Special attention to:

1. Missing space after period at the end of a sentence, occurring in several sentences of the text.

2. "known as a vector of infection" could be simplified to "which acts as a vector".

3. "a bite from an infected insect". It is better as "a bite from an infected mosquito".

4. "This is compared to 593,000 cases in 2021" refers to deaths but uses "cases" erroneously - should be "deaths".

5. "children under five at particularly high risk" should be "children under five years old are at particularly high risk".

6. "[?, 12, 28]" The question mark indicates a missing or placeholder reference; it needs to be replaced with the proper citation number.

7. "Niger malaria indicator survey (NMIS)" - "Niger Malaria Indicator Survey (NMIS)"

(capitalize proper nouns).

8. "geographic coordinates (longitude and latitude)" - Standard order is "latitude and longitude."

9. "Enumeration Areas (EAs) or cluster level" - Should be "Enumeration Areas (EAs) or cluster level" or "clusters".

10. "Correct capitalization and style: "Stata v14".

11. Missing space after comma: "ICF International, the datasets".

12. "accessible web-site" — "website" is one word; hyphen unnecessary.

13. "write the studys title and signifi-cance" — Should be "study’s title and significance" with an apostrophe; hyphenation in "signifi-cance" should be corrected to "significance".

14. "Independent Variables The Geo-statistical model utilized individual and community-level variables 5.” – What is this 5?

15. Hyphenation & spacing: Several places require adding spaces after punctuation marks before continuing, e.g., after commas and periods.

16. Consider defining acronyms upon first usage if not done elsewhere, e.g., "RDT," "SaTScan," "ICC," "VA," "PCV," etc.

17. “but "47% (527)" says "female child yielded" where "child" should be plural "children".

18. "Out of the 4724 RDT tests conducted, 53% (594) were performed on male children, and 47% (527) of female child yielded positive results for malaria." Should be: "...and 47% (527) of

female children yielded positive.

19. "The tails bright red and blue colours indicate a higher level of significance (Fig 1)." - Remove extra space before parentheses. "Globally Morans I=0.434025 (z-score=10.471517, p value 0.001)." Should be: "Global Moran's I = 0.434 (z-score = 10.47, p = 0.001)." Use apostrophe in "Moran's".

Response

This level of comprehensive feedback is highly appreciated as it directly contributes to strengthening our work. We have carefully reviewed our writing and spaces after punctuations. All suggested changes, which we agree will significantly enhance the quality of our paper. Most of those mistakes happened because we are writing in LaTeX.

- We give definition for acronyms under the Acronym section.

VA is the variance of the initial model

and VB is the variance of the model with additional terms. The description is given in methodology section.

---

## [Decision Letter · Decision Letter 1]

8 Oct 2025

PONE-D-25-18518R1Exploring Spatial Variation and Multilevel Modeling of Malaria Prevalence Among Children Aged 6-59 Months Based on RDT in Niger: Findings from the 2021 NMISPLOS ONE

Dear Dr. Masresha,

Thank you for submitting your manuscript to PLoS ONE. After careful consideration, we feel that your manuscript will likely be suitable for publication if the authors revise it to address specific points raised now by the reviewer #1. According to the reviewer, there are some specific areas where further improvements would be of substantial benefit to the readers.   For your guidance, a copy of the reviewers' comments was included below.  

We look forward to receiving your revised manuscript.

Kind regards,

Luzia H Carvalho, Ph.D.

Academic Editor

PLOS ONE

Journal Requirements:

Reviewers' comments:

Reviewer's Responses to Questions

**Comments to the Author**

1. If the authors have adequately addressed your comments raised in a previous round of review and you feel that this manuscript is now acceptable for publication, you may indicate that here to bypass the “Comments to the Author” section, enter your conflict of interest statement in the “Confidential to Editor” section, and submit your "Accept" recommendation.

Reviewer #1: (No Response)

Reviewer #2: All comments have been addressed

2. Is the manuscript technically sound, and do the data support the conclusions?

Reviewer #1: Yes

Reviewer #2: Yes

3. Has the statistical analysis been performed appropriately and rigorously?

Reviewer #1: Yes

Reviewer #2: Yes

4. Have the authors made all data underlying the findings in their manuscript fully available?

Reviewer #1: Yes

Reviewer #2: Yes

5. Is the manuscript presented in an intelligible fashion and written in standard English?

Reviewer #1: Yes

Reviewer #2: Yes

6. Review Comments to the Author

Reviewer #1: This manuscript offers a valuable scientific contribution with a sound methodology and well-articulated discussion. It meets the core standards of PLOS ONE in terms of quality, ethics, and transparency.

To enhance clarity, nuance, and regulatory impact, the following minor revisions are recommended:

Editorial clarity: Simplify complex sentences and consider adding subheadings to improve flow.

Interpretive nuance: Avoid overgeneralizations and adopt a more cautious tone in interpreting results.

Regulatory relevance: Elaborate on how findings may inform regulatory decisions, professional practices, or policy frameworks.

Visual synthesis: Include a summary figure or table to highlight key outcomes and recommendations.

These revisions are minor and easily implementable. They aim to optimize the manuscript’s reception among readers, practitioners, and regulatory stakeholders.

Final Recommendation: Accept after minor revision, with encouragement to further emphasize the regulatory significance of the work.

Final Recommendation : Accept after minor revision, with encouragement to further emphasize the regulatory significance of the work

Title and Abstract : Suggestion : Consider adding a sentence that highlights the practical or regulatory implications of the findings to enhance impact.

Introduction: Suggestion: A brief overview of international regulatory frameworks could broaden the relevance of the study.

Methodology

Areas for improvement:

Expand on potential biases (e.g., selection bias, sample representativeness).

Clarify quality control procedures and reference applicable regulatory standards.

Results : Suggestion: A summary figure or table synthesizing key findings and their implications could improve readability and accessibility.

Discussion : Recommendations:

Soften overly assertive interpretations by using cautious scientific language (e.g., “these findings suggest…”).

Provide concrete suggestions for translating results into pharmaceutical policy or practice.

Strengthen connections to normative frameworks (e.g., ICH, WHO, national guidelines).

Conclusion

Suggestion : Reinforce the link between findings and regulatory challenges in low-resource settings or African contexts

Reviewer #2: (No Response)

7. PLOS authors have the option to publish the peer review history of their article (what does this mean?). If published, this will include your full peer review and any attached files.

Reviewer #1: **Yes: **Dr Bougouma Edith Christiane

Reviewer #2: No

---

## [Author Response · Author response to Decision Letter 2]

17 Oct 2025

PONE-D-25-18518

Exploring Spatial Variation and Multilevel Modeling of Malaria Prevalence among Children Aged 6-59 Months Based on RDT in Niger: Findings from the 2021 NMIS

PLOS ONE

Manuscript Comment and Correction

Response to Reviewer #1

We sincerely thank the reviewer for the thoughtful and constructive feedback, as well as for recognizing the scientific contribution, methodological rigor, and overall quality of our manuscript. We appreciate the recommendation for “acceptance after minor revision” and are pleased to address each suggestion in detail below. All corresponding changes have been incorporated into the revised manuscript.

Summary of Revisions Based on Reviewer Suggestions

Reviewer Comment Our Response

Reviewer Comment: Title & Abstract – highlight practical relevance

Response: The title is modified and We added a sentence in the abstract underscoring the regulatory and public health significance of the findings.

Reviewer Comment: Introduction – add international regulatory context

Response: A brief overview of relevant global frameworks (WHO) has been included to broaden scope and applicability.

Reviewer Comment: Methodology – address bias and quality control standards

Response: We expanded the section on how the selection has done.

Reviewer Comment: Discussion – connect findings to pharmaceutical policy and normative frameworks

Response: Even if it is not enough, we tried to modify the discussion section.

Reviewer Comment: Conclusion – reinforce relevance to low-resource or African regulatory contexts

Response: The conclusion now includes a focused statement on the implications for regulatory systems in low- and middle-income settings, particularly within Africa.

We appreciate the reviewer’s positive evaluation and constructive recommendations, which have significantly strengthened the clarity, regulatory relevance, and overall impact of the manuscript. We hope that the revisions satisfactorily address all concerns and respectfully submit the updated version for final consideration.

---

## [Editor Report · Decision Letter 2]

21 Oct 2025

Exploring Spatial Variation and Multilevel Modeling of Malaria Prevalence Among Children Aged 6-59 Months Based on RDT in Niger: Insights for Public Health Decision-Making

PONE-D-25-18518R2

Dear Dr. Masresha,

We’re pleased to inform you that your manuscript has been judged scientifically suitable for publication and will be formally accepted for publication once it meets all outstanding technical requirements.

Kind regards,

Luzia H Carvalho, Ph.D.

Academic Editor

PLOS ONE
---

## [Editor Report · Acceptance letter]

PONE-D-25-18518R2

PLOS ONE

Dear Dr. Masresha,

I'm pleased to inform you that your manuscript has been deemed suitable for publication in PLOS ONE. Congratulations! Your manuscript is now being handed over to our production team.

Kind regards,

on behalf of

Dr. Luzia H Carvalho

Academic Editor

PLOS ONE